# ByLimb: Development of a New Technique to Implant Intracorneal Ring-Segments from the Perilimbal Region

**DOI:** 10.3390/life13061283

**Published:** 2023-05-30

**Authors:** Roberto Albertazzi, Roger Zaldivar, Carlos Rocha-de-Lossada

**Affiliations:** 1Centro de Ojos Quilmes, Quilmes 1865, Argentina; 2Instituto Zaldivar, Ciudad de Buenos Aires 1865, Argentina; zaldivarroger@gmail.com; 3Department of Ophthalmology, Qvision, VITHAS Almería Hospital, 04120 Almeria, Spain; 4Department of Ophthalmology, Regional Universitary Hospital of Málaga, 18014 Granada, Spain; 5Department of Surgery, Ophthalmology Area, University of Seville, 41012 Seville, Spain; 6Department of Opthalmology, Vithas Malaga, 29016 Malaga, Spain

**Keywords:** keratoconus, ICRS, femtosecond laser, surgical technique, corneal biomechanics

## Abstract

A new technique that allows implanting intracorneal ring-segments (ICRS) from the limbal zone is described. Using a femtosecond laser (FSL), a 360° corneal tunnel is created with an internal diameter of 5.4 mm and an external diameter of 7.0 mm, with a wider area (0.2 mm inner and 0.2 mm outer) in the upper 60° of the tunnel (called landing zone). Next, a 4.36 mm-long corneal-limbal incision was created with the FSL, which connects to the bubbles created in the landing zone. The entire procedure was performed using intraoperative optical coherence tomography (OCT). Once the two incisions were connected using blunt-edged Mac Pherson forceps, the bubbles were released from the surgical plane. The programmed ICRS(s), 6 mm in diameter, are then placed in the corneal tunnel from the limbal incision with the aid of Sinskey forceps. Finally, when the ICRS is in place, the surgery is complete.

## 1. Introduction

Different designs of intracorneal segment designs (ICRS) have been developed. They have been shown to be useful in the therapeutic management of keratoconus and secondary ectasia [1,2]. In turn, the femtosecond laser (FSL), with the assistance of intraoperative optical coherence tomography (OCT), contributed to improving aspects related to the reproducibility and effectiveness of the technique [3]. However, it is worth mentioning that the implantation of ICRS following the traditional approach could lead to some potential corneal wound healing problems due to the alteration created in the incision area of the epithelium and the roof of the corneal tunnel. This altered corneal epithelium can subsequently develop into some complications, such as corneal stromal melting, corneal infections, and even extrusion of the ICRS [4,5].

Although the FSL-assisted technique has a low percentage of intra- and post-operative complications [6], our group hypothesized that it might be possible to develop a new ICRS implantation technique that creates an external incision of the tunnel, located near the sclero-corneal limbus, thus obtaining a hermetic intracorneal tunnel that could reduce problems related to wound healing. The aim of this article is to describe the development of a technique to implant ICRS from the perilimbal region without affecting the roof of the corneal tunnel.

## 2. Methods

After conducting experimental proofs of concept through computational simulations and surgeries on cadaveric pig eyes, a phased research plan for a pilot study was designed to evaluate this new technique in the eyes of patients with keratoconus. The research protocol was evaluated and approved by the ethics committee of the Argentine Council of Ophthalmology. The patients gave their informed consent after receiving a detailed explanation of the characteristics of the procedure under study to be performed. Among the inclusion criteria, patients with keratoconus grade 1, 2, or 3 following Amsler classification who had no contraindications to FSL-assisted ICRS implant surgery were selected. Although this technique was developed in order to have the possibility of implanting ICRS of 6 mm diameter of different commercial brands with different terminations (straight and rounded) and different angles (90°, 120°, and 150°), in this preliminary study we used Ferrara’s ICRS (Ferrara Ophthalmics, Belo Horizonte, Brazil).

To assess wound healing at the implantation site, observation and follow-up were performed by slit lamp and also by OCT images (Optovue Avanti; Optovue Inc., Fremont, CA, USA), 24 h and one week after surgery.

### 2.1. Surgical Technique Description

#### 2.1.1. Preparation of the Surgical Site

Antisepsis was performed on periocular skin, including eyelids, eyebrows, and the conjunctival sac, using 0.5% povidone iodine, and a sterile self-adhesive field and blepharostat were placed.

#### 2.1.2. First Docking, Creation of the Corneal Tunnel

A first docking was performed using the LDV Z8 FSL (Ziemer, Bern, Switzerland) in order to create a 360° closed corneal tunnel with an internal diameter of 5.4 mm and an external diameter of 7.0 mm. The depth of the tunnel was programmed from the thinnest corneal pachymetry at 70 μm from the endothelium, whose thinnest part was evaluated by intraoperative corneal OCT.

Landing zone: term used to denote a zone located in the upper 60° of the tunnel, where the tunnel was enlarged (0.2 mm inner and 0.2 mm outer: 0.4 mm additional in total), as shown in Figure 1A.

#### 2.1.3. Second Docking, Limbal Incision Creation

A second docking was performed, and the FSL created a limbal incision area with a length of 4.36 mm. Its depth was determined by observing the intraoperative OCT until reaching the denominated landing zone, which can be identified and visualized by the fusion of the bubbles originating within the previously created corneal tunnel, as shown in Figure 1B.

#### 2.1.4. Connection of the Limbal Incision with the Corneal Tunnel

The limbal incision was opened with blunt-edged Mac Pherson forceps, releasing the bubbles and trying to find the surgical plane in order to connect the limbal incision area with the corneal tunnel area, creating a route to enter the tunnel.

#### 2.1.5. ICRS Implantation

Then the segments were placed, only with the help of a Sinskey hook, directly inside the tunnel, positioning the ICRS in the correct place. The Sinskey hook was removed, and the surgery was completed.

At the end of the surgery, two drops of topical Gatifloxacin 0.5% (Zymaxid^®^Allergan, Zymaxid, Allergan Inc., Irvine, CA, USA) were placed, and the patient was instructed to use them 3 times a day for 7 days.

## 3. Results

The technique described has been implemented without the occurrence of intraoperative complications in 17 eyes. The sequence of a surgery is presented in Figure 2 and Appendix A.

In relation to wound healing, no early postoperative problems related to tunnel epithelialization, ICRS migration/extrusion, or inflammatory disorders have been observed in the 17 eyes included in this study. The postoperative appearance observed by slit lamp at 24 h and after the first week is presented in Figure 3 and Figure 4, respectively. No postoperative complications have been observed in any of the eyes over three months of follow-up (these patients are still being evaluated in a study that has a follow-up of 1 year).

Figure 5 shows the slit-lamp images from two different eyes. The illumination is directed to the perilimbal area, where a slight haze is observed at the wound healing edge one week after surgery. In the same images, the ICRS can also be seen properly placed in the corneal tunnel with no signs of inflammation.

A sequence of corneal OCT images shows the perilimbal incision area and the tunnel area where the ICRS is implanted (Figure 6). A thin wound healing line can be seen progressing from the epithelium to the stromal tunnel one week after surgery. The edges of the wound are closed, as is the so-called “landing zone”. The OCT also allows us to observe the integrity of the corneal tunnel roof.

## 4. Discussion

In this work, an FSL-assisted technique has been presented, which offers the possibility of performing ICRS implantation through a limbal incision. Until now, regardless of the ICRS model used, all the techniques described in the literature, whether manual or FSL-assisted, required performing implantation through at least one corneal incision, penetrating the roof of the tunnel, with the ends of the segments remaining close to the incision. This means that in these surgeries, it is sometimes necessary to close the corneal wound with a 9/0 or 10/0 nylon suture at the end of the surgery, especially when the volume of the implants impedes the correct apposition of the edges of the wound. Although this approach to implanting ICRS, especially with the assistance of FSL, is effective and safe, it is not without potential complications [3,6,7]. The most frequent problems are related to the insertion zone of the ICRS, where there can be seeding of the corneal epithelium with subsequent localized corneal melting and migration of the segment, finally causing the extrusion of the implant in some cases [7,8]. Whenever any incision on the channel in its roof is performed with any of the techniques described (manual or FSL assisted), this incision impedes surgeons from placing an ICRS not only below it but also at 10° and 15° from it because the loss of continuity that originates this incision facilitates the introduction of the epithelium into the canal, thus increasing the rate of complications previously mentioned.

Through this new technique, it is possible to create a 360° closed tunnel, giving us the possibility to innovate by exploring new ICRS designs such as concatenated combinations of profiles within the same channel or injectable or articulated segments. In addition, it may be possible to use and experiment with different materials, whose flexibility and design allow us to improve the personalization of treatments. Among the advantages of this new way of implanting ICRS, it is proposed that by avoiding affecting the roof of the corneal tunnel, the ends of the implants are farther away from the incision. Our hypothesis is that performing this new technique, where the upper wall of the corneal tunnel (roof) is completely undamaged, could reduce the possibility of contact of the implants in the corneal healing zone, as happens in traditional techniques, whether manual or FSL-assisted, providing greater protection to the implanted prosthesis and subsequently reducing the potential associated corneal wound healing complications. It should be noted that the area of the incision coincides with the region of Vogt’s palisade, which provides an earlier closure [9], similar to what occurs in the wounds performed in phacoemulsification cataract surgeries. Likely, another potential advantage of this technique is that if for any reason it is necessary to reposition the ICRS and it is performed in an earlier or postoperative period, which is relatively simple to access from the limbal incision to the channel through the corneal-limbal wound, being very similar to what happens when it is necessary to lift the flap of a laser in situ keratomileusis (LASIK) if necessary. On the other hand, performing this new technique requires more time since two docking stages assisted by the FSL must be conducted. An interesting aspect is that the bubbles generated during the first docking are useful to clearly define and visualize by OCT the area to create the landing zone in the limbal incision developed by the second docking, as we described earlier in the methods section.

Likewise, just as OCT has been disruptive in the diagnosis and follow-up of retinal pathologies, it is clear that nowadays it is essential for corneal evaluation [10]. Intraoperative OCT is also essential to perform this surgical technique and to use the same imaging system for postoperative follow-up. In this way, it is possible to have a better observation of the perilimbal incision area, the so-called “landing zone”, and also to evaluate and follow the evolution of each sector of the corneal tunnel.

This technique has been developed to be feasible for the implantation of any of the commercially available ICRS on the market, with different profiles, arches (90°–120°–140° and 150°), and ends, all of them with a 6 mm diameter. We consider that this approach is another access route to performing the ICRS implant, which can allow further improvement of the safety and efficacy results of this type of corneal implant. At the same time, this technique is versatile enough to be adapted to new concepts of corneal implants. In relation to the level of difficulty, the learning curve for its implementation, in principle, does not pose a major surgical skill challenge for an anterior segment surgeon. In addition, this new technique could allow for the rearrangement of the ICRS according to unexpected outcomes related to the wrong keratometric axis or rotation. Moreover, it could allow for an easier post-operative replacement of the ICRS if necessary in order to achieve better post-operative refractive outcomes. In addition, this approach can facilitate the use of the 360° of the tunnel if considered by the surgeon (i.e., 4 × 90° each or 3 × 120° each).

Although the clinical aspect of immediate corneal healing seems to be optimal and no complications have been found in 17 operated eyes, the long-term safety of this technique is currently being evaluated in a pilot study with a 1-year follow-up in the eyes of keratoconus patients implanted with Ferrara ICRS. It would also be interesting to have studies on animal models in order to better understand aspects of corneal healing in relation to the effect of femtosecond laser in the perilimbal area described in this work, as well as to know the intrastromal reaction of ICRS implanted by this technique and its reversibility, as previously studied with the traditional technique by Ibares-Frías et al. [11,12].

Finally, we hope that in a short time, the first results of the effectiveness and safety of this technique can be presented. In conclusion, this study has presented a new technique assisted by FSL that offers the possibility of performing the implantation of ICRS through a perilimbal incision. Its potential advantages should be clinically demonstrated in future studies.

## Figures and Tables

**Figure 1 life-13-01283-f001:**
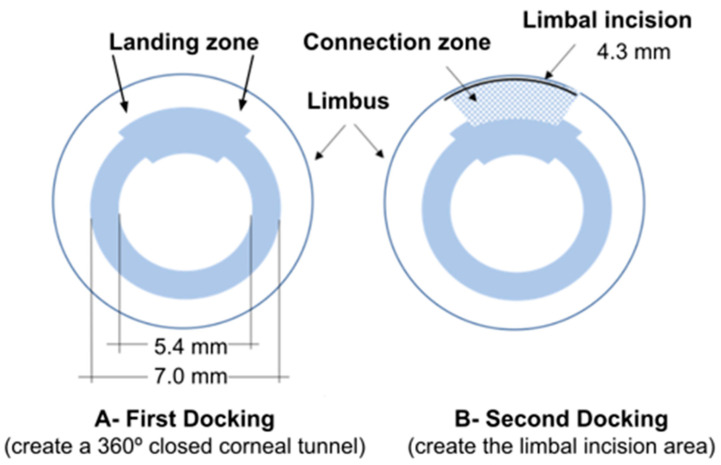
Schematics representing the corneal tunnel, with a wider area at the superior level (part **A**) and the limbal incision (part **B**).

**Figure 2 life-13-01283-f002:**
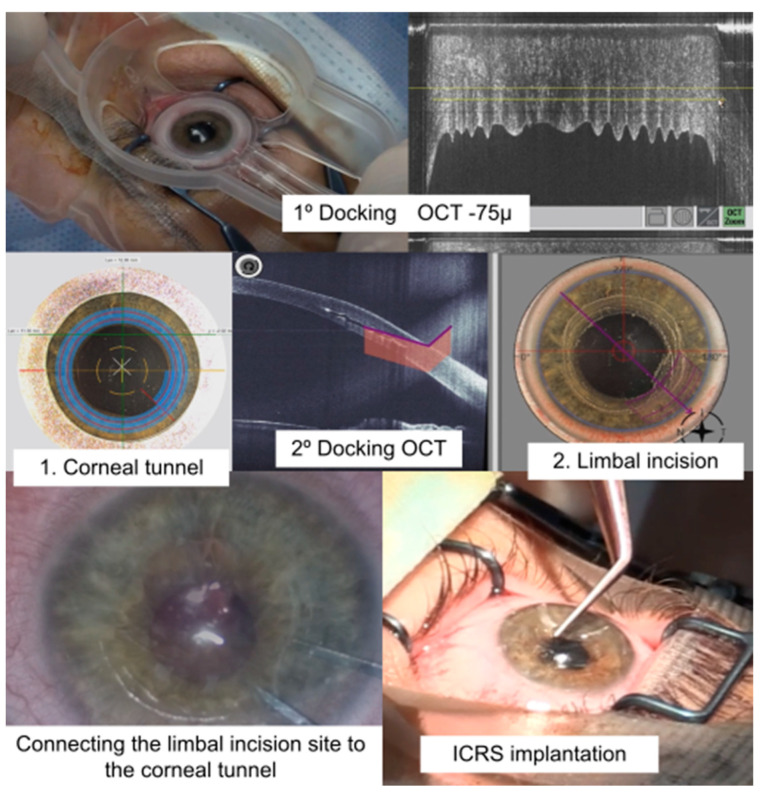
Stages of limbal ICRS implantation: the first and second docking can be observed together with the creation of the connection of the limbal area with the corneal tunnel and the subsequent implantation of the ICRS. Intraoperative anterior segment OCT images, slit lamp pictures of the patient’s eye, together with illustrative diagrams are also shown.

**Figure 3 life-13-01283-f003:**
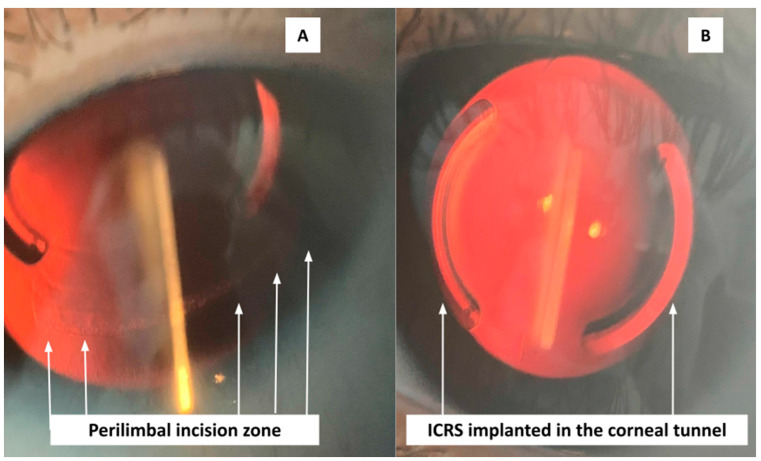
Slit lamp aspect of the perilimbal incision area (**A**) and ICRS implanted in the corneal tunnel (**B**), 24 h after surgery.

**Figure 4 life-13-01283-f004:**
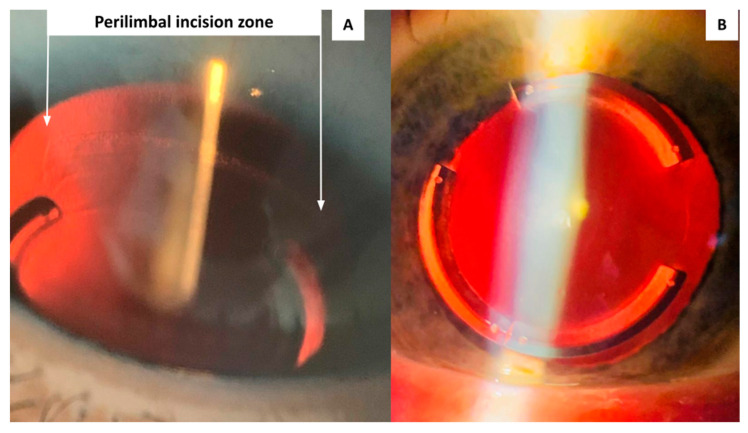
Slit lamp aspect of the perilimbal incision area (**A**) and ICRS implanted in the corneal tunnel (**B**), one week after surgery.

**Figure 5 life-13-01283-f005:**
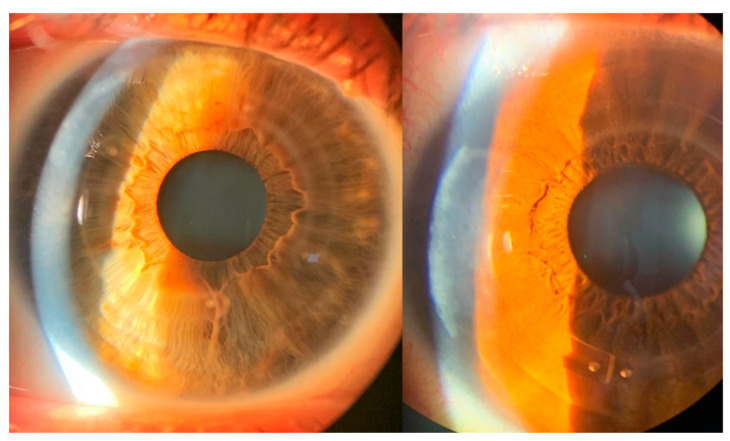
Slit-lamp images of two different cases show the edge of the incision site (directly illuminated by the slit lamp). Both eyes also show the implanted ICRS, with no signs of corneal inflammation.

**Figure 6 life-13-01283-f006:**
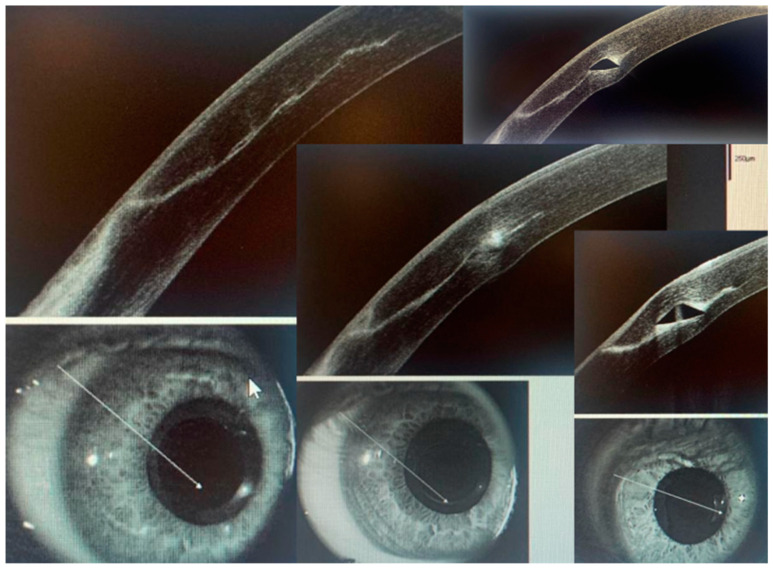
Different corneal OCT images are presented to observe the wound healing zone, from the epithelium to the stromal tunnel where the ICRS is placed.

## Data Availability

The data presented in this study are available upon request from the corresponding author.

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
