# Peer review of "ByLimb: Development of a New Technique to Implant Intracorneal Ring-Segments from the Perilimbal Region"

_life, 2023, doi:10.3390/life13061283_

Round 1

Reviewer 1 Report

Dear authors, the work you have done is very interesting. You have developed a new way of implanting ICRS, which potentially seems to have advantages over the current procedure. 

The abstract, the introduction, and the rest of the development of the paper is very good. The discussion highlights the importance of the work.

In your manuscript, methodologically, the aim you propose, to describe a new technique, is fulfilled. Your conclusions are related to the objective.

It is understood that this work is mainly to share a new technique and make it available to the rest of the community. It is understood that you will have to wait to complete the finalization of the results in order to be able to evaluate efficacy and safety in the medium and long term. But we hope that the publication of this technique may motivate other surgeons to consider this possible surgical method that protects the roof of the canal.

As you state in the discussion, there will be many future studies that should evaluate this technique, including basic science studies, to demonstrate its benefits, which theoretically seem to be clear.

In conclusion, you have developed and described a new way of implanting ICRS, which is likely to gain many followers in the future.

Congratulations and let's move forward

Author Response

Thank you for your comments.

We hope that we can move forward and that other groups will be able to reproduce and evaluate this technique.

Reviewer 2 Report

Excellent new technique; congrats. However, i have some minor issues I would like to expose

1) Introduction, page 2, lines 33-34:

Corneal ring complications with FSL are not usually seen, which should be described more clearly. I would emphasize the new possibilities with this new tunnel creation rather than avoid complications.

2) Methods

a) Minor English revision

b) line 57 --> Ferrara ICRS needs to have a company brand, city and country (Ferrara Ophthalmics, Belo Horizonte, Brazil)

c) Since you are describing a new technique, I would appreciate it if you mention how many eyes have been operated on until now.

3) Discussion

a) minor English revision

b) In the discussion, you should present the rate of complications you cited (based on the literature) and discuss these rates.

c) Even though this is not a clinical trial, it is recommendable to describe if you had any complication as migration, dislocation, inflammation, melting, wound healing difficulty or loss of vision lines. And if you had any extra difficulty with the extra docking necessary to perform this technique.

Minor English revision especially in methods and discussion

Author Response

Reviewer #2

Excellent new technique; congrats. However, i have some minor issues I would like to expose

In the new version of the manuscript, we have made the English edition and we have made the modifications that you have suggested.

Thank you very much for your contribution and help us improve.

1) Introduction, page 2, lines 33-34:

Corneal ring complications with FSL are not usually seen, which should be described more clearly. I would emphasize the new possibilities with this new tunnel creation rather than avoid complications.

Considering your suggestion, it was modified.

2) Methods

a) Minor English revision.

Thank you. We have done a second edition of English with an external consultant.

b) line 57 --> Ferrara ICRS needs to have a company brand, city and country (Ferrara Ophthalmics, Belo Horizonte, Brazil).

It was included.

c) Since you are describing a new technique, I would appreciate it if you mention how many eyes have been operated on until now.

Yes, 17 eyes until now. These patients are under follow-up. We have included that information in “results” section.

3) Discussion

a) minor English revision. Thank you. Our new version was revised and improved.

b) In the discussion, you should present the rate of complications you cited (based on the literature) and discuss these rates.

*We understand that reporting this new technique generates the desire to know all the details. But the objective of this work is to describe the development of a new technique, and in this way, to put it into the consideration of the scientific community. But your comment is correct and we have included more information in the results section, with the data we have with the 3 months follow-up of the 17 operated eyes.
**The study in progress has a final follow-up of one year, where we will be able to better analyze final aspects of safety and efficacy. However, in view of the good initial results and having observed that there have been no intraoperative or postoperative complications. We have the series of patients with different follow-up times and so far, no complications have been observed. In fact, some patients are close to the end of the study, with good data in relation to efficacy.
***We hope soon to have all the data of the series with one year of follow-up, to be able to make this report, but given the originality of this technique, and considering that its growth will also depend on the opinion and improvements that other groups can make, we wish to put it to the consideration of future readers.

****According to your comments, we have also made a change in the “discussion”, addressing the aspect of surgical and postoperative complications.

c) Even though this is not a clinical trial, it is recommendable to describe if you had any complication as migration, dislocation, inflammation, melting, wound healing difficulty or loss of vision lines. And if you had any extra difficulty with the extra docking necessary to perform this technique.

The response to this comment was described above, in point (b). However, there is something that might be interesting to share, regarding the topic of additional coupling. Initially, when we were developing this technique, we also theoretically considered that the extra coupling could be an "extra difficulty" for the surgeon. However, the bubbles generated by the first docking, observed by the OCT images, help us to clearly see and define the "landing zone", as described in the Methods section; Point 3.

But thanks to your comments, we have added something in the manuscript (at the end of the second paragraph of the discussion).

Reviewer 3 Report

1. Line 71 Is that 70"um"?

2. Only 24 hours and 1 week results were present. How is the long-term efficacy and safty?

3. If the patients need a keratoplasty, the limbal tunnel will be a trouble.

Moderate editing of English language

Author Response

Rewier #3

1. Line 71 Is that 70"um"?

Yes, it was corrected

2. Only 24 hours and 1 week results were present. How is the long-term efficacy and safty?

In the current work, we wish to report the development of this technique and its characteristics; this work is part of an ongoing study with a follow-up at 1 year and monthly controls during the first six months and quarterly, up to one year. Currently, 17 eyes have been operated and no intraoperative or postoperative complications have been observed. We have case-by-case efficacy data and hope to be able to report the entire series, possibly by the end of this year. Beyond that, we wish to share the characteristics of this technique with the scientific community, considering that it is something original so that other groups can consider it but at the same time criticize it and potentially improve it.

3. If the patients need a keratoplasty, the limbal tunnel will be a trouble.

Thank you for your comments. We estimate that there should be no problem if tomorrow any of these patients require penetrating keratoplasty (PK). The diameter used for the first 360° docking, where the ICRS are implanted, is 7.0 mm, as shown in Figure 1. Most PKs are programmed from 7.5 mm. In relation to the limbal area, although it exceeds this limit, it is a femtosecond programmed incision similar to the one that could be programmed to perform femtosecond laser-assisted cataract surgery (FLACS).
This limbal incision, through which the ICRS are introduced, subsequently heals, as in FLACS.
Therefore, we estimate that if a patient requires a PK in the future, it would be similar to a patient who has had previous FLACS surgery.

Round 2

Reviewer 3 Report

Thank you for your reply